# Blood Basophils Relevance in Chronic Rhinosinusitis with Aspirin-Exacerbated Respiratory Disease

**DOI:** 10.3390/diagnostics13111920

**Published:** 2023-05-31

**Authors:** Giuseppe Brescia, Cristoforo Fabbris, Leonardo Calvanese, Luigia Bandolin, Barbara Pedruzzi, Valerio Maria Di Pasquale Fiasca, Silvia Marciani, Francesca Mularoni, Fabio Degli Esposti Pallotti, Michael Negrisolo, Giacomo Spinato, Anna Chiara Frigo, Gino Marioni

**Affiliations:** 1ENT Unit, Department of Surgery, Ospedali Riuniti Padova Sud, 35043 Padova, Italy; 2Department of Medicine DIMED, Padova University, 35100 Padova, Italy; 3Department of Neuroscience DNS, Section of Otolaryngology, Padova University, 35100 Padova, Italy; 4Department of Neuroscience DNS, Padova University, 35100 Padova, Italy; 5Department of Cardiac-Thoracic-Vascular Sciences and Public Health, Padova University, 35100 Padova, Italy; 6Phoniatrics and Audiology Unit, Department of Neuroscience DNS, Padova University, 31100 Treviso, Italy

**Keywords:** AERD, CRSwNP, nasal polyposis, basophils, eosinophils, blood cell count

## Abstract

Aspirin-exacerbated respiratory disease (AERD) is characterized by eosinophilic asthma, chronic rhinosinusitis with nasal polyps (CRSwNP) and intolerance to cyclooxygenase-1 inhibitors. Interest is emerging in studying the role of circulating inflammatory cells in CRSwNP pathogenesis and its course, as well as their potential use for a patient-tailored approach. By releasing IL-4, basophils play a crucial role in activating the Th2-mediated response. The main aim of this study was to, first, investigate the level of the pre-operative blood basophils’ values, blood basophil/lymphocyte ratio (bBLR) and blood eosinophil-to-basophil ratio (bEBR) as predictors of recurrent polyps after endoscopic sinus surgery (ESS) in AERD patients. The secondary aim was to compare the blood basophil-related variables of the AERD series (study group) with those of a control group of 95 consecutive cases of histologically non-eosinophilic CRSwNP. The AERD group showed a higher recurrence rate than the control group (*p* < 0.0001). The pre-operative blood basophil count and pre-operative bEBR were higher in AERD patients than in the control group (*p* = 0.0364 and *p* = 0.0006, respectively). The results of this study support the hypothesis that polyps removal may contribute to reducing the inflammation and activation of basophils.

## 1. Introduction

Aspirin-exacerbated respiratory disease (AERD) is an inflammatory condition that consists of eosinophilic asthma, chronic rhinosinusitis with nasal polyps (CRSwNP) and respiratory reactions to cyclooxygenase-1 (COX-1) inhibitors. The reported prevalence of AERD in CRSwNP patients ranges between 8.7% and 26% [1,2,3,4,5]. Compared with asthmatics, AERD patients may be predominantly characterized by severe asthma according to the American Thoracic Society/European Respiratory Society definition (which includes patients with refractory asthma and those in whom treatment of comorbidities, such as severe sinus disease remains incomplete [2,6,7]) and by aggressive CRSwNP in terms of a higher probability of recurrence after surgery [2,3,8].

A standard histological examination can identify the main inflammatory cells (such as eosinophils and neutrophils) that have been associated with CRSwNP relapse [9,10,11,12]. Recently, attention has turned to the value of blood sampling and inflammatory cell assays to shed light on the pathophysiology of CRSwNP and predict the course of the disease. Using minimally invasive methods (such as blood sampling), these conventional cytological parameters could make it easier to (i) provide patients with appropriate information pre- and post-operatively, (ii) adopt rational follow-up protocols and (iii) administer dedicated post-operative medical treatments to patients at high risk of recurrence [10,13,14,15]. A direct association between CRSwNP recurrence rates and blood eosinophil and basophil counts was reported. In 2017, our clinical research group preliminarily aimed to (i) identify the best-fitting cutoffs for binarizing pre-operative blood eosinophils and basophils (counts/percentages) for prognostic purposes in cases of CRSwNP recurrence after surgery and (ii) distinguish said cutoffs for prognosticating the recurrence in patients with histologically diagnosed eosinophilic vs. non-eosinophilic CRSwNP [16].

In 2004, Hartnell et al. [17] investigated the mechanisms behind basophil recruitment from blood into tissues in human nasal polyp extracts. Given the emerging role of basophils in CRSwNP, the prognostic value of the blood basophil/lymphocyte ratio (bBLR) was tested in a large CRSwNP series; the bBLR was significantly higher in patients whose disease recurred than in those who remained recurrence-free [18]. In 2017, a retrospective study was performed on 334 patients with CRSwNP to compare the preoperative blood eosinophil-to-basophil ratio (bEBR) between different endotypes and with controls (69 cases) [19]. The mean bEBR was significantly higher in the CRSwNP group than in the control group with no evidence of nasal, paranasal or systemic inflammatory disorders. Furthermore, the mean bEBR was significantly higher in the CRSwNP sub-cohorts with allergies, asthma and AERD.

It is unknown which biomarkers effectively predict the recurrence of nasal polyposis in patients with AERD undergoing endoscopic sinus surgery (ESS). The main aim of this study was to, first, investigate the level of blood basophils, bBLR and bEBR as predictors of recurrent polyps after the ESS in AERD patients. The secondary aim was to compare the blood basophil-related variables of the AERD series (study group) with those of a control group of 95 consecutive cases of histologically non-eosinophilic CRSwNP.

## 2. Materials and Methods

### 2.1. Patients

This retrospective study was conducted in accordance with the principles of the Helsinki Declaration. All patients signed a detailed informed consent form and gave their written permission for clinical case publication. The data were examined in agreement with the Italian privacy and sensitive data laws, as well as the internal regulations of Padova University’s Otolaryngology Section. Furthermore, all enrolled patients signed a form in which they consented “to the use of their clinical data for scientific research purposes in the medical, biomedical and epidemiological fields, also in order to be recalled in the future for follow-up needs”.

A consecutive series of 39 adult patients who had undergone surgery for CRSwNP with AERD at the Otolaryngology Section of Padova University between 2009 and 2022 was retrospectively assessed. In contrast, the control group consisted of another consecutive series of 95 CRS cases with a histological diagnosis of non-eosinophilic polyposis who had undergone surgery by the same surgical team in the same period. Patients were excluded from the study in the event of a lack of pre-operatory blood basophils data, a diagnosis of systemic inflammatory disease or autoimmune disease, or acute or chronic infectious conditions other than sinusitis; cancer; blood disorders; a history of systemic corticosteroid use; or chronic kidney disease. Patients whose CRSwNP had already recurred before their blood was sampled post-operatively were also excluded, as their recurrent disease might have influenced their post-ESS eosinophil, basophil and lymphocyte counts.

Information about sensitivity to acetylsalicylic acid or other non-steroidal anti-inflammatory drugs (NSAIDs) was obtained from their medical history recorded in Padova University Hospital’s electronic archives (Galileo). Preoperative blood cell counts and percentages were obtained for each patient. The necessary laboratory tests were performed at least three months after withdrawing oral steroids and one month after withdrawing nasal steroids; they were all processed at the same laboratory (Laboratory Medicine Service, Padova University Hospital) and certified in compliance with ISO standard 15,189. The total and specific IgE for Dermatophagoides pteronyssinus and Dermatophagoides farinae, birch pollen, pellitory, grass mix, cat and dog dander, Alternaria alternata, Aspergillus fumigatus and common ragweed were determined. A diagnosis of asthma was confirmed according to the definition of the Global Initiative on Asthma [20]. The recruitment process is summarized in Figure 1.

After the ESS, surgical tissue was stained with hematoxylin and eosin to measure the eosinophil count, examining 5 high-power fields (HPFs) (400×) selected from each sample and recording the average number of eosinophils. The eosinophilic histotype corresponded to a mean score of ≥10 eosinophils/HPF [21]. Patients with a mean tissue eosinophil count of <10 were considered to be histologically non-eosinophilic (control group of the present investigation).

All patients were treated post-operatively with isotonic saline solution irrigations twice a day (20 mL per irrigation), nasal steroids (fluticasone furoate 110 μg daily (55 μg per nostril) or mometasone furoate 200 μg daily (100 μg per nostril)). Adequate therapy was prescribed for asthmatic and allergic patients. Follow-ups with rigid 0° or 30° endoscopes were scheduled 3, 6 and 12 months after the ESS, and yearly thereafter.

From a laboratory viewpoint, for all patients considered in the study, the following pre- and postoperative variables were determined and archived: blood basophils count (cells × 10^9^/L), blood basophils percentage, bBLR and bEBR.

### 2.2. Statistical Methods

The statistical analysis was performed with SAS 9.4 for Windows (SAS Institute Inc., Cary, NC, USA). The results are reported as the median and range for quantitative variables and as the count and percentage for categorical variables.

The recurrence-free interval was measured as the time from the ESS to recurrence or the last follow-up evaluation for censored patients.

The prognostic role of each blood basophils-related variable on recurrence-free interval for AERD and control patients was assessed with univariate Cox regression.

The plot of the cumulative Martingale residuals against the values of the covariate and Kolmogorov-type supremum test based on a sample of 1000 simulated residual patterns was used to test the proportionality for quantitative covariates. The results are expressed as the *p*-value and hazard ratio (HR) with a 95% confidence interval (CI).

The AERD and control groups were compared with the Mann–Whitney test for quantitative variables after having checked the normality with the Shapiro–Wilk test and the Q–Q plot. The chi-square test was applied for categorical variables.

A *p*-value < 0.05 was considered indicative of statistical significance.

## 3. Results

A total of 39 patients with AERD (19 females and 20 males, mean age 52.7 years, range 26–81 years) and 95 consecutive control cases with non-eosinophilic CRSwNP (37 females and 58 males, mean age 51.4 years, range 24–80 years) were enrolled in the study.

### 3.1. Inter-Group Analysis (AERD Group vs. Control Group)

The CRSwNP recurrence rate after ESS was significantly higher in the AERD group than in the control group, with a calculated HR of 5.473 (95% CI 2.753–10.881) (Table 1).

By analyzing the pre-operative blood basophil counts, the AERD group’s median value was found to be significantly higher than that of the control group (0.03 [range 0.01–0.11] vs. 0.03 [0.01–0.08] cells × 10^9^/L, respectively; *p*-value 0.0364) (Figure 2).

Interestingly, considering the same variable post-operatively, there was no significant difference between the two groups (*p*-value 0.4931).

Furthermore, the median value of the pre-operative eosinophils/basophils ratio was significantly higher in the AERD group than in the control group (11.75 [range 0.00–49.00] vs. 7.00 [range 0.57–29.00], respectively; *p*-value 0.0006) (Figure 3).

Furthermore, this basophil-related variable was not significantly different between the two cohorts after surgery (*p*-value 0.6607).

Table 2 reports in detail the results of the inter-group analysis (AERD vs. control group).

### 3.2. Intra-Group Analysis

When considering the AERD group, no significant differences were found when comparing the blood basophil-related variables regarding the recurrence rate of CRSwNP after the ESS (Table 3).

Moreover, in the control group of histologically non-eosinophilic CRSwNP, pre- and post-operative blood basophil-related values were not predictive of the nasal polyps recurrence rate (Table 4).

## 4. Discussion

There is definitely an emerging interest in investigating the pathophysiological role of basophils in CRSwNP [22,23,24]. Basophils are the rarest granulocytes, but recent studies have demonstrated that they have a crucial, non-redundant role in the immune system. Basophils release IL-4 in both an IgE-dependent and IgE-independent manner in response to a variety of stimuli. It is widely accepted that IL-4 plays a major part in promoting the differentiation of naïve CD4+ T cells into Th2 cells. Basophil-derived IL-4 enhances the expressions of IL-5, IL-13, IL-33 and γ-interferon in type 2 innate lymphoid cells, leading to the accumulation of eosinophils [16,25,26,27,28,29,30].

It was found that basophils can promote Th2 polarization or the activation of Th2 cells when exposed to haptens and peptide antigens. It is worth noting that basophils can also promote Th2 polarization in the presence of dendritic cells [31]. Human Treg cells not only suppress but rather activate basophils and promote Th2 responses through IL-3-dependent and signal-transducer-and-activator-of-transcription-5-dependent mechanisms [32,33]. However, further research is mandatory to fully understand the functions of basophils and their role in the immune response.

Patients with CRSwNP characterized by a type 2 immune signature often have severe and recurrent disease. When CRSwNP is accompanied by comorbid asthma, it is associated with more severe sinus symptoms; worse quality of life; and is more difficult to treat, both medically and surgically. AERD is an endotype of this combination. Individuals with AERD tend to have the most severe form of the disease and resist treatment [3]. AERD is characterized by high levels of type 2 innate immune cells, such as mast cells, eosinophils and basophils, with significant increases in the levels of traditional type 2 inflammatory mediators, such as interleukins (ILs) 4, 5 and 13 and eotaxins 1 and 2 [34,35,36,37]. According to our study, a significantly higher number of relapses after surgery occurred among AERD patients compared with the controls. This confirmed a worse course of the disease among the former.

Regarding patients with AERD, basophils were found to be hyperactivated within the polyps [10,38]. ESS enables clearance of the polyps and polypoid mucosa, removal of the inflammatory tissue and a reduction of the load of antigens that trigger the inflammation [39]. Consistent with the abovementioned biological mechanism, an ESS could correspond to a reduction in the blood eosinophil count in CRSwNP [40]. In the present investigation, pre-operative blood basophil counts and bEBR were significantly higher in AERD than in the histologically non-eosinophil CRSwNP controls. After the ESS, these two values were not significantly different between the AERD subjects and controls, and this result might be explained by a decrease in inflammation and by the low number of available cases. No other significant differences in blood count values emerged from the inter-group and intra-group analyses, either pre- or post-operatively. Moreover, in the AERD group, a rather large inter-individual difference was found in the pre-operative blood basophil counts. This great difference could be due to the fact that type Th2 inflammation can be started by numerous mediators [31]; therefore, basophil counts could be raised only in some cases. Further studies in larger cohorts are needed in order to give an explanation.

Basophil-derived IL-4 enhances the expressions of IL-5, IL-9 and IL-13 in type 2 innate lymphoid cells, as well as the chemokine CCL11 (better known as eosinophil chemotactic protein, or eotaxin-1), leading to an accumulation of eosinophils [41,42]. New classes of biological drugs that block the production or action of Th2-related cytokines (IL-4, IL-5, IL-13) are making important progress toward new therapeutic paradigms in CRSwNP and its endotypes [37]. The response to biological drugs that are prescribed exclusively for Th2-related polyposis could be monitored clinically and by measuring blood eosinophilia and basophilia [43]. According to our results, no post-operative significant differences were noted in blood basophilia between the two groups, whereas this difference was significant before surgery. This may lead to hypothesizing a possible role of surgery, even if it was not the purpose of this study. Moreover, a possible combined effect of surgery and biological therapy should be investigated with further studies.

The main weaknesses of this study concern the retrospective design and the limited number of patients, in particular those with available post-operative laboratory data. The main strengths lie in the intra-group homogeneity of the series of patients considered since the (i) histopathological analyses were all done by a dedicated head and neck pathologist; (ii) ESS was performed by the same team of surgeons; (iii) the endoscopic follow-up after surgery was conducted by the same team; (iv) recurrent eCRSwNP was always confirmed endoscopically; and, lastly, (v) all blood tests were performed over a long-term post-operative period at the same laboratory.

## 5. Conclusions

Before surgery, among subjects with CRSwNP, blood basophil counts and bEBR were significantly higher in patients with AERD than in those without AERD. The retrospective nature of the study did not allow for correlating these data with significant blood increases of any cytokines involved. AERD was associated with a higher risk of recurrence of nasal polyps. Among all the CRSwNP cases, polyps removal with ESS may have helped in reducing inflammation and the activation of basophils and eosinophils. The same result is obtained with biological therapy. Therefore, our observations seemed interesting in relation to the routine use of anti-interleukin biologics. Furthermore, a possible synergistic effect of surgery associated with the biologic therapy should be further investigated.

## Figures and Tables

**Figure 1 diagnostics-13-01920-f001:**
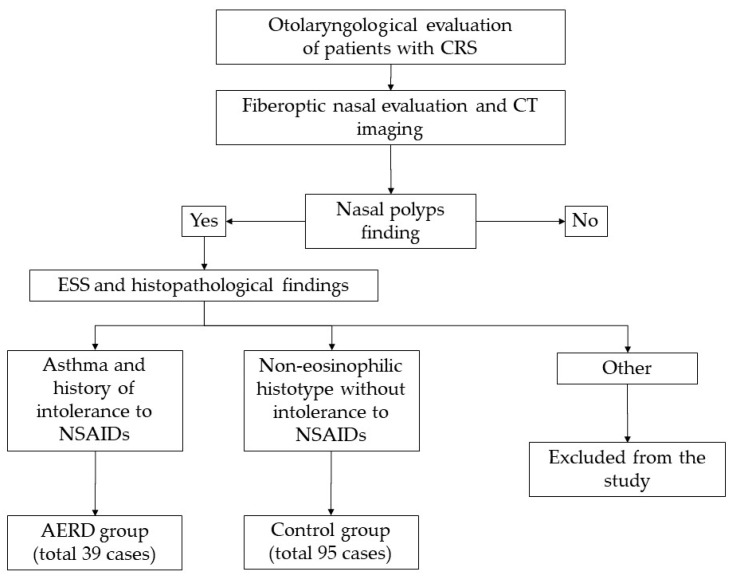
Recruitment process of CRSwNP patients enrolled in the study. Abbreviations: CRS, chronic rhinosinusitis; CT, computed tomography; ESS, endoscopic sinus surgery; NSAIDs, non-steroidal anti-inflammatory drugs; AERD, aspirin-exacerbated respiratory disease.

**Figure 2 diagnostics-13-01920-f002:**
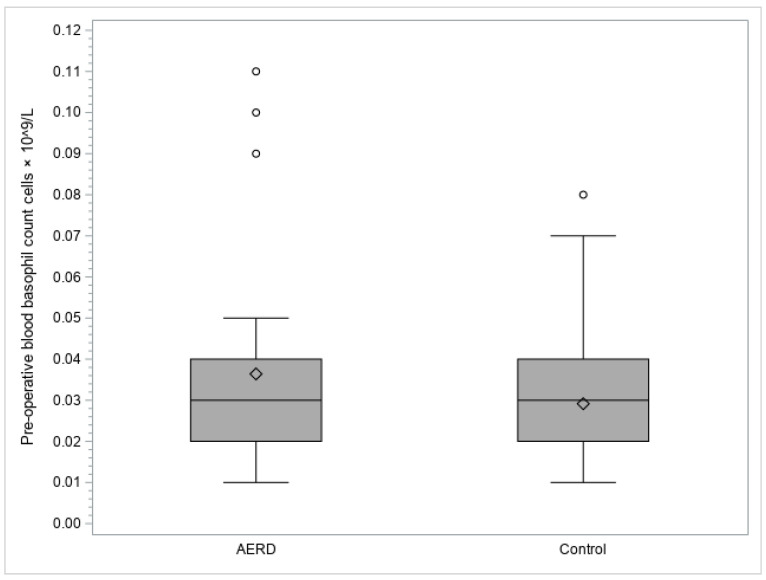
Box plot representation of the pre-operative blood basophil counts (cells × 10^9^/L). The top and bottom edges of each box indicate the interquartile range; the line that runs through each box represents the median; the diamond in each box represents the mean; the whiskers that extend from each box cover the extent of the data less than or equal to 1.5 times the interquartile range; circles represent the outlier values. Abbreviation: AERD, aspirin-exacerbated respiratory disease.

**Figure 3 diagnostics-13-01920-f003:**
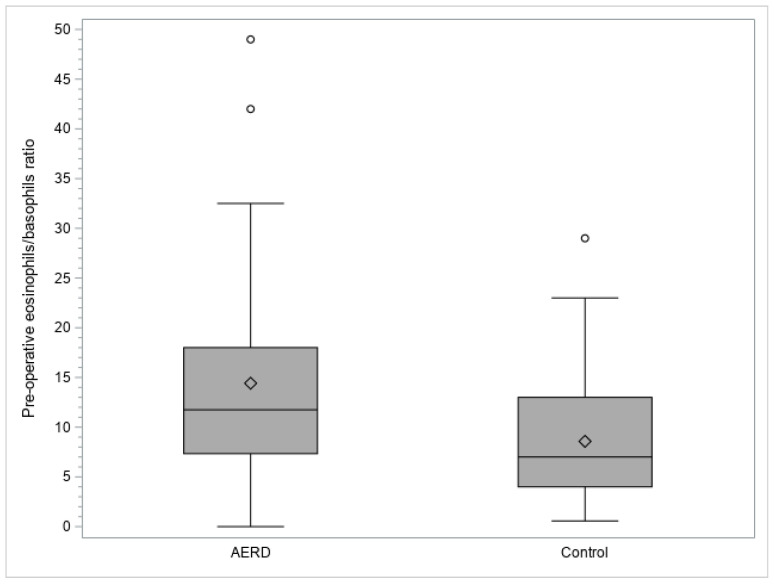
Box plot representation of the pre-operative eosinophils/basophils ratios. The top and bottom edges of each box indicate the interquartile range; the line that runs through each box represents the median; the diamond in each box represents the mean; the whiskers that extend from each box cover the extent of the data less than or equal to 1.5 times the interquartile range; circles represent the outlier values. Abbreviation: AERD, aspirin-exacerbated respiratory disease.

**Table 1 diagnostics-13-01920-t001:** AERD group vs. control group: analysis of CRSwNP recurrences.

	Recurrence			
	No (N = 100)	Yes (N = 34)	*p*-Value	HR (95% CI)
**Group**				
**N missing**	0	0	<0.0001	
**Control group**	81 (81.0%)	14 (41.2%)		1
**AERD group**	19 (19.0%)	20 (58.8%)		5.473 (2.753; 10.881)

N—number of cases, HR—hazard ratio, CI—confidence interval.

**Table 2 diagnostics-13-01920-t002:** Laboratory data comparison between the AERD and control groups. N—number of cases.

	Group		
	AERD (N = 39)	Control (N = 95)	*p*-Value
Gender			
N missing	0	0	0.2976
Male	20 (51.3%)	58 (61.1%)	
Female	19 (48.7%)	37 (38.9%)	
Age			
N missing	0	0	
Median (range)	50.0 (26.0–81.0)	52.0 (24.0–80.0)	0.6842
Pre-operative blood basophil count (cells × 10^9^/L)			
N missing	0	0	
Median (range)	0.03 (0.01–0.11)	0.03 (0.01–0.08)	0.0364
Pre-operative blood basophils rate			
N missing	0	0	
Median (range)	0.50 (0.10–1.90)	0.40 (0.10–1.50)	0.0640
Pre-operative basophils/lymphocytes ratio			
N missing	3	20	
Median (range)	0.02 (0.00–0.10)	0.01 (0.00–0.07)	0.1322
Post-operative blood basophil count (cells × 10^9^/L)			
N missing	15	0	
Median (range)	0.03 (0.00–0.40)	0.03 (0.00–0.11)	0.4931
Post-operative blood basophils rate			
N missing	15	0	
Median (range)	0.40 (0.00–6.50)	0.40 (0.00–1.90)	0.4301
Post-operative basophils/lymphocytes ratio			
N missing	19	20	
Median (range)	0.01 (0.01–0.39)	0.01 (0.00–0.07)	0.4061
Pre-operative eosinophils/basophils ratio			
N missing	0	0	
Median (range)	11.75 (0.00–49.00)	7.00 (0.57–29.00)	0.0006
Post-operative eosinophils/basophils ratio			
N missing	16	1	
Median (range)	8.20 (0.00–28.00)	7.00 (0.50–316.67)	0.6607

**Table 3 diagnostics-13-01920-t003:** Analysis of CRSwNP recurrences within the AERD group.

	Recurrence			
	No (N = 19)	Yes (N = 20)	*p*-Value	HR (95% CI)
Gender				
N missing	00 (00.0%)	00 (00.0%)		
Male	11 (57.9%)	09 (45.0%)	0.5093	1
Female	08 (42.1%)	11 (55.0%)		1.346 (0.557; 3.251)
Age				
N missing	0	0		
			0.2768	0.982 (0.950; 1.015)
Median (range)	52.00 (39.00–81.00)	47.50 (26.00–75.00)		
Pre-operative blood basophil count (cells × 10^9^/L)				
N missing	19 (0)	0		
			0.6817	39.568 (0.000; 1.7004 × 10^9^)
Median (range)	0.03 (0.01–0.10)	0.04 (0.01–0.11)		
Pre-operative blood basophils rate				
N missing	0	0		
			0.6398	1.259 (0.480; 3.304)
Median (range)	0.50 (0.10–1.50)	0.55 (0.20–1.90)		
Pre-operative basophils/lymphocytes ratio				
N missing	2	1		
			0.6469	55.564 (0.000; 1.6246 × 10^9^)
Median (range)	0.02 (0.00–0.03)	0.02 (0.00–0.10)		
Post-operative blood basophil count (cells × 10^9^/L)				
N missing	7	8		
			0.8425	0.575 (0.002; 135.804)
Median (range)	0.03 (0.00–0.20)	0.03 (0.01–0.40)		
Post-operative blood basophils rate				
N missing	7	8		
			0.4250	1.137 (0.829; 1.561)
Median (range)	0.45 (0.00–0.90)	0.40 (0.10–6.50)		
Post-operative basophils/lymphocytes ratio				
N missing	10	9		
			0.9205	1.299 (0.008; 221.911)
Median (range)	0.01 (0.01–0.03)	0.02 (0.01–0.39)		
Pre-operative eosinophils/basophils ratio				
N missing	0	0		
			0.8419	0.996 (0.953; 1.040)
Median (range)	12.75 (2.40–32.50)	10.78 (0.00–49.00)		
Post-operative eosinophils/basophils ratio				
N missing	8	8		
			0.3505	0.967 (0.900; 1.038)
Median (range)	8.20 (0.20–28.00)	8.75 (0.00–18.00)		

N—number of cases, HR—hazard ratio, CI—confidence interval.

**Table 4 diagnostics-13-01920-t004:** Analysis of CRSwNP recurrences within the control group.

	Recurrence			
	No (N = 81)	Yes (N = 14)	*p*-Value	HR (95% CI)
Gender				
N missing	0	0		
Male	47 (58.0%)	11 (78.6%)		1
Female	34 (42.0%)	03 (21.4%)	0.1161	0.353 (0.097; 1.293)
Age				
N missing	0	0		
			0.1445	0.973 (0.939; 1.009)
Median (range)	55.00 (25.00–80.00)	43.50 (24.00–75.00)		
Pre-operative blood basophil count (cells × 10^9^/L)				
N missing	0	0		
			0.7505	84.067 (0.000; 6.133 × 10^13^)
Median (range)	0.03 (0.01–0.08)	0.02 (0.01–0.07)		
Pre-operative blood basophils rate				
N missing	0	0		
			0.8598	1.172 (0.201; 6.850)
Median (range)	0.40 (0.10–1.50)	0.30 (0.10–1.30)		
Pre-operative basophils/lymphocytes ratio				
N missing	19	1		
			0.7319	1620.023 (0.000; 3.696 × 10^21^)
Median (range)	0.01 (0.00–0.07)	0.02 (0.01–0.04)		
Post-operative blood basophil count				
N missing	0	0		
			0.6936	0.005 (0.000; 1.867 × 10^9^)
Median (range)	0.03 (0.00–0.11)	0.03 (0.01–0.06)		
Post-operative blood basophils rate (cells × 10^9^/L)				
N missing	0	0		
			0.6521	0.672 (0.119; 3.785)
Median (range)	0.40 (0.00–1.90)	0.40 (0.20–0.80)		
Post-operative basophils/lymphocytes ratio				
N missing	19	1		
			0.7757	0.002 (0.000; 1.229 × 10^16^)
Median (range)	0.01 (0.00–0.07)	0.02 (0.01–0.03)		
Pre-operative eosinophils/basophils ratio				
N missing	0	0		
			0.6218	1.021 (0.940; 1.108)
Median (range)	7.00 (0.57–29.00)	7.11 (1.00–23.00)		
Post-operative eosinophils/basophils ratio				
N missing	1	0		
			0.9939	1.000 (0.977; 1.023)
Median (range)	6.71 (0.50–316.67)	11.00 (0.50–21.50)		

N—number of cases, HR—hazard ratio, CI—confidence interval.

## Data Availability

The datasets generated and analyzed during the study are available upon reasonable request.

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
