# Peer review of "Blood Basophils Relevance in Chronic Rhinosinusitis with Aspirin-Exacerbated Respiratory Disease"

_diagnostics, 2023, doi:10.3390/diagnostics13111920_

Round 1

Reviewer 1 Report

Thank you  for  the opportunity to review this manuscript.

In order  to improve its methodological  quality,  some  mayor  concerns  must be  taken into account:

State  if  preoperative basophils or basophils/lymphocyte ratio  correlates with  clinical   data (i.e. a  surrogate marker  of  severity)

Add a figure  describing  the patient recruitment process.

State how  you assessed the  normality of  data.

State, if  possible, if  any other  variable  correlated  with  the  risk of  recurrence  and  if a  clinical score  may  be  feasible.

Tables are  not clear: include  only  pertinent central tendency/dispersion  values.

Author Response

Dear Reviewer,

thank you for your comments. Please find the following answers to your questions.

Thank you for the opportunity to review this manuscript. In order to improve its methodological quality, some mayor concerns must be taken into account:

  • State if preoperative basophils or basophils/lymphocyte ratio correlates with clinical   data (i.e. a surrogate marker of severity)

► Thanks for your suggestion. Since its establishment in 2015, our research group on CRSwNP has preferred to refer only to objective and non-subjective clinical, laboratory, pathological (conventional, structural histopathological, immunohistochemical) and prognostic variables. Also, in this study the aims focus only on objective variables which are specifically demographic, laboratory and prognostic (in terms of recurrence of the endoscopically diagnosed nasal polyps).

  • Add a figure describing the patient recruitment process.

►As suggested, the new Figure 1(Methods section) summarizes the patient recruitment process.

  • State how you assessed the normality of data.

►Following reviewer suggestion, we reported that the normality of data was assessed with Shapiro-Wilk test and Q-Q plot.

  • State, if possible, if any other variable correlated with the risk of recurrence and if a clinical score may be feasible.

►Thanks for your comment. The study design was mainly to evaluate the prognostic ability of variables related to circulating inflammatory cells in the AERD case study group (Table 3) and also in the control group (Table 4). The other variables considered (demographic) were not associated with the recurrence event in either group (see Tables 3 and 4). Furthermore, the numerical limits of the study group in particular do not allow the problem to be approached by trying to create a demographic/laboratory score.

  • Tables are not clear: include only pertinent central tendency/dispersion values.

► According to the suggestion of the referee, we dropped the mean and standard deviation in the modified Tables.

Reviewer 2 Report

I appreciate the opportunity to review the manuscript for publication in MDPI Diagnosis. I feel that the topics are interesting, however, the manuscript is grossly premature which should be modified.

I have a few comments as follows.

The authors examined the roles of blood basophils by using bBLR and bEBR, as predictors of recurrent polyps after ESS in AERD patients.

In Figure 1, I found rather large inter-individual differences in pre-operative blood basophil counts from the AERD group. The authors should describe and explain possible reasons and causative factors.

The authors show basophil-relate variables turned to be not significantly different between the two cohorts after surgery.

This might be simply due to small in numbers (missing cases).

L212: “Consistent with the above-mentioned biological mechanism, ESS could correspond to a reduction in the blood eosinophil count in CRSwNP [42].”

The authors should demonstrate in this article.

I am curious about subset of lymphocytes and changes in related arachidonic acid metabolites.

Above all, IL-4 level measurement in serum is necessary based o the authors’ hypothesis.

Author Response

Dear Reviewer,

thank you for your comments. Please find the following answers to your questions.

  • I appreciate the opportunity to review the manuscript for publication in MDPI Diagnosis. I feel that the topics are interesting, however, the manuscript is grossly premature which should be modified. I have a few comments as follows. The authors examined the roles of blood basophils by using bBLR and bEBR, as predictors of recurrent polyps after ESS in AERD patients. In Figure 1, I found rather large inter-individual differences in pre-operative blood basophil counts from the AERD group. The authors should describe and explain possible reasons and causative factors.

►Thank you for your note. We re-evaluated our database and patients’ charts, but nor clinical findings or history data could explain this result. In this limited number of cases, the great difference could be due to the fact that type Th2 inflammation can be started by numerous mediators, therefore basophils could be raised only in some cases. Furthermore, not all AERD causes are known, and the pathogenesis is not fully understood, so there may be some differences in the development of this disease that could lead to these differences.

We added a paragraph in the Discussion to respond to your comment.

  • The authors show basophil-relate variables turned to be not significantly different between the two cohorts after surgery. This might be simply due to small in numbers (missing cases).

► Thank you for your comment. We agree with your punctuation and added a sentence to the Discussion.

  • L212: “Consistent with the above-mentioned biological mechanism, ESS could correspond to a reduction in the blood eosinophil count in CRSwNP [42].” The authors should demonstrate in this article.

 ►Thank you for your comment. The sentence refers to one of our previous articles (cited) where we were able to analyze the reduction of eosinophils in the blood after ESS. As you pointed out in the previous comment, unfortunately this evidence is not clearly demonstrable in the present study given the smaller number of cases. We added a sentence in the Discussion.

  • I am curious about subset of lymphocytes and changes in related arachidonic acid metabolites.

Above all, IL-4 level measurement in serum is necessary based on the authors’ hypothesis.

►Thank you for your comment. Your request is surely interesting and could add important information. On the other hand, IL-4 level measurement is not routinely performed. Since ours is a retrospective study, its design was based only on available laboratory data.

Reviewer 3 Report

This is a nice research article about an important clinical problem. The paper is well written and quite easy to understand. There are only minor corrections to be made, like in line 96 (check the citation), Tab. 1 (P value is missing in the AERD group) or Tab 3 (check and correct 95%CI values, esp. xxxEyy). There is one question I would like to address to the authors. Did the tolerance to aspirin or other NSAIDs change after surgery? I am wondering if the patients were more prone to respiratory infections after surgery? (This is only my curiosity, not necessarily to be added into the text)

Author Response

Dear Reviewer,

thank you for your comments. Please find our answers to your comments.

  • This is a nice research article about an important clinical problem. The paper is well written and quite easy to understand. There are only minor corrections to be made, like in line 96 (check the citation)

►Thank you for your comment. “Galileo” is the name of electronic archives in our Hospital. We changed brackets in order to avoid misunderstanding.

  • Tab. 1 (P value is missing in the AERD group)

►The reported p-value was calculated comparing the two groups in terms of recurrence rate.

  • or Tab 3 (check and correct 95%CI values, esp. xxxEyy).

►Thank you for your comment. We changed the notation in the tables.

  • There is one question I would like to address to the authors. Did the tolerance to aspirin or other NSAIDs change after surgery? I am wondering if the patients were more prone to respiratory infections after surgery? (This is only my curiosity, not necessarily to be added into the text)

►This is an interesting question. We cannot evaluate this aspect in this retrospective setting. It could be studied in a further investigation.

Round 2

Reviewer 1 Report

Corrections  are OK,  but  the sended  document  has erasures/overscripts that needs to be  erased.

Author Response

Thanks for your comment. The corrections you requested have been made.

Reviewer 2 Report

I appreciate the opportunity to review again the manuscript for publication in MDPI Diagnostics.

I reckon that the manuscript has been revised and improved in part in accordance with the reviewers’ comments. However, the amendments in the revised manuscript still fail to support the highlights of the article. No additional data have been presented.

Author Response

The present study showed significantly higher levels of blood basophil count and eosinophil-to-basophil ratio among patients with AERD. Unfortunately, the retrospective nature of the study did not allow correlating these findings with significant blood differences of any of the cytokines involved. Therefore, no further data could be provided. However, our observations also seemed potentially interesting in relation to the use of anti-interleukins biological drugs.

We have changed the Conclusions section to clarify this point and replaced the term "role" of basophils with "level" of basophils in the abstract and Introduction sections.

Round 3

Reviewer 2 Report

I understand the situation and defense comments.